# Integrative genomic analysis of the human immune response to influenza vaccination

Luis M Franco[1,2†], Kristine L Bucasas[1†], Janet M Wells[3], Diane Niño[3], Xueqing Wang[4], Gladys E Zapata[4], Nancy Arden[5], Alexander Renwick[6], Peng Yu[1], John M Quarles[5], Molly S Bray[4‡], Robert B Couch[2,3], John W Belmont[1,4*], Chad A Shaw[1]

[1]Department of Molecular and Human Genetics, Baylor College of Medicine, Houston, United States; [2]Department of Medicine, Baylor College of Medicine, Houston, United States; [3]Department of Molecular Virology and Microbiology, Baylor College of Medicine, Houston, United States; [4]Children's Nutrition Research Center, Baylor College of Medicine, Houston, United States; [5]Department of Microbial and Molecular Pathogenesis, Texas A&M University System Health Science Center, College Station, United States; [6]Interdepartmental Program in Structural and Computational Biology and Molecular Biophysics, Baylor College of Medicine, Houston, United States

**Abstract** Identification of the host genetic factors that contribute to variation in vaccine responsiveness may uncover important mechanisms affecting vaccine efficacy. We carried out an integrative, longitudinal study combining genetic, transcriptional, and immunologic data in humans given seasonal influenza vaccine. We identified 20 genes exhibiting a transcriptional response to vaccination, significant genotype effects on gene expression, and correlation between the transcriptional and antibody responses. The results show that variation at the level of genes involved in membrane trafficking and antigen processing significantly influences the human response to influenza vaccination. More broadly, we demonstrate that an integrative study design is an efficient alternative to existing methods for the identification of genes involved in complex traits.

*For correspondence: jbelmont@bcm.edu

†These authors contributed equally to this work

Present address: ‡School of Public Health, University of Alabama at Birmingham, Birmingham, United States

Competing interests: The authors declare that no competing interests exist.

## Introduction

Influenza remains one of the major threats to human health worldwide and is responsible for an estimated 250,000–500,000 deaths each year (*World Health Organization, 2009*). Attempts at immunization pre-dated the isolation of the virus from humans in 1933 (*Smith et al., 1933*) and vaccination remains the cornerstone of prevention strategies. Since 1977, strains of influenza A (H3N2), influenza A (H1N1), and influenza B have been responsible for the majority of documented human infections and trivalent vaccines are updated annually to contain the circulating strains. Animal models have demonstrated that immune responses and susceptibility to influenza infection can be strongly influenced by host genetic factors (*Trammell and Toth, 2008*; *Srivastava et al., 2009*). As with viral infection, variability in the immune response to vaccination is likely to be influenced by genotype. Accordingly, twin and sibling studies have shown heritability estimates as high as 45% for a varicella vaccine (*Klein et al., 2007*) and 90% for a measles vaccine (*Tan et al., 2001*). Studies investigating influenza vaccine immunogenicity in humans have consistently shown large inter-individual variability, but the genetic contribution to this variability remains poorly understood.

**eLife digest** Vaccines increase resistance to disease by priming the immune system to respond to specific viruses or microorganisms. By presenting a weakened (or dead) form of a pathogen, or its toxins or surface proteins, to the immune system, vaccines trigger the production of antibodies against the virus or microorganism. If a vaccinated individual then encounters the pathogen, their immune system should be able to recognize and destroy it. Many vaccines also include a secondary agent, known as an adjuvant, to further stimulate the immune response.

Influenza, an RNA virus commonly referred to as the 'flu', is an infectious disease that affects both birds and mammals. Seasonal epidemics occur each year affecting 2–7% of the population. According to the World Health Organization, influenza leads to nearly 5 million hospitalizations each year and causes up to half a million deaths. Vaccination is a primary strategy for the prevention of seasonal influenza, but responses to the vaccine vary markedly, partly because of variation in the genetic makeup or genotype of individuals. However, the details of how genes influence response to vaccination, and indeed susceptibility to influenza, remain unclear.

To investigate the genetic basis of variation in the immune response of healthy adults to the seasonal influenza vaccine, Franco et al. combined information about the genotypes of individuals with measurements of their gene transcription and antibody response to vaccination. They identified 20 genes that contributed to differential immune responses to the vaccine. Almost half of these encode proteins that are not specifically associated with the immune system, but have more general roles in processes such as membrane trafficking and intracellular transport.

Focusing on these genes may enable researchers to spot those individuals who are less likely to respond to a vaccine. It could also open up new avenues of research for vaccine development: rather than designing adjuvants that target known immune mechanisms, researchers should develop adjuvants that target the proteins encoded by these 20 genes.

Gene expression is strongly controlled by common genetic variants (*Morley et al., 2004*; *Stranger et al., 2007*) with both broad (*Bullaughey et al., 2009*) and tissue-specific effects (*Innocenti et al., 2011*; *Rotival et al., 2011*), referred to as expression quantitative trait loci (eQTL). Moreover, genome-wide association studies have identified hundreds of variants associated with human disease risk that are also eQTL, implying that the mechanism by which they influence risk involves variation in transcriptional responses (*Emilsson et al., 2008*; *Cookson et al., 2009*; *Naukkarinen et al., 2010*; *Nicolae et al., 2010*; *Rotival et al., 2011*; *Barreiro et al., 2012*) Finally, integrative genomic studies in model organisms (*Schadt et al., 2005*; *Amit et al., 2009*) have demonstrated that the combination of genetic and transcriptional information can allow direct tests of causal mechanisms in controlled experiments. We hypothesized that integrating genome-wide genotype data with serial measurements of the transcriptional and humoral responses to an influenza vaccine in a clinical study could be used to identify loci that influence vaccine responsiveness and subsequent immunity to influenza in humans.

We immunized an ethnically homogeneous group of 119 healthy adult male volunteers with licensed trivalent influenza vaccine. DNA was obtained from peripheral blood and genome-wide SNP genotyping was performed. We also measured global transcript abundance in peripheral blood RNA specimens before and at three time points (days 1, 3, and 14) after vaccination. Type-specific antibody measurements (H1N1, H3N2, and FluB) were made in serum samples before and at two time points (days 14 and 28) after vaccination. An identical study was then carried out with an independent validation cohort of 128 ethnically homogeneous healthy adult female volunteers. This experimental design allowed us to search for loci that show evidence of a transcriptional response to vaccination, genetic regulation of gene expression (*cis*-acting eQTL), and correlation between gene expression and the magnitude of the antibody response.

## Results

### Multiple genes show evidence of a transcriptional response to the vaccine and genetic regulation of expression

We performed mixed model regression analysis with SNPs located in 1-Mb intervals around each expression reporter sequence. We began by identifying SNP-transcript pairs with both significant evidence of a *cis*-acting eQTL and significant changes in gene expression in response to vaccination. Thresholds for local significance were initially explored, since only SNPs flanking each reporter sequence were tested for *cis* association. In the discovery cohort, 3229 SNP-transcript pairs, corresponding to 408 unique genes, exhibited significant genotype-expression association (genotype effect $p < 1 \times 10^{-4}$) and concomitant evidence of a transcriptional response to the vaccine (day effect $p < 0.01$). Of these, 2606 SNP-transcript pairs, corresponding to 256 genes, were validated in the independent cohort of female volunteers (genotype effect $p < 0.05$ and day effect $p < 0.01$). When more stringent thresholds were applied, 756 SNP-transcript pairs, corresponding to 114 unique genes, exhibited significant genotype-expression association (genotype effect $p < 5 \times 10^{-8}$) and concomitant evidence of a transcriptional response to the vaccine (day effect $p < 0.01$) in the discovery cohort. Of these, 654 SNP-transcript pairs, corresponding to 93 genes, were validated in the second cohort (genotype effect $p < 0.05$ and day effect $p < 0.01$). A majority of these (467 SNP-transcript pairs, corresponding to 78 unique genes) would pass equally stringent thresholds in both cohorts (genotype effect $p < 5 \times 10^{-8}$, day effect $p < 0.01$). A Manhattan plot of these results is presented in *Figure 1*. Data for the individual SNP-transcript pairs that passed equally stringent thresholds in both cohorts, including results of significance testing and gene identifiers, are provided in Table 1 via the Interactive Results Tool (which is also available to download from Zenodo and shown within *Supplementary file 1*).

### At some loci, the genetic effect is enhanced or only apparent after the experimental perturbation

We hypothesized that, at some loci, the magnitude of the genetic effect could be different before and at different time points after vaccination. This type of effect, which would not be observed in a cross-sectional study design, could be directly examined with our serial expression data. We analyzed the additive effect of genotype on expression at each day in the study. Using a *cis*-effect significance threshold of $p < 1 \times 10^{-4}$ in the discovery cohort and $p < 0.05$ in the validation cohort, this analysis identified 5155 validated eQTL SNP-transcript pairs (3011 at baseline and 3417, 2496, and 3043 at days 1, 3, and 14, respectively). These SNP-transcript pairs correspond to 543 unique genes. We then identified the SNP-transcript pairs in which the expression variance explained was most strongly increased after vaccination (highest change in genetic variance explained, which we termed delta-$R_g^2$). This analysis revealed multiple loci at which the genetic effect was either enhanced or only apparent after the experimental perturbation. An example is presented in *Figure 2A*, which displays local Manhattan plots for the *NECAB2* locus before and 3 days after vaccination in both cohorts.

Theoretically, the observed temporal changes in the estimated genotype effect after vaccination could be driven by an increase in the effect size, a relative decrease in the variability within genotype strata, or both. We analyzed all SNP-transcript pairs for loci at which we observed both a strong *cis*-acting eQTL and a transcriptional response to vaccination, calculating the relative magnitude of slope and within-genotype variance between the pre-vaccination and maximal $R_g^2$ time points. *Figure 2B* shows that an increase in the strength of the genotype effect (slope of the additive association) was the main driver for the observed change in $R_g^2$, and that this amplitude change was a general feature of the loci in which we observed both a strong *cis*-acting eQTL and a transcriptional response to the vaccine stimulus. The delta-$R_g^2$ values were consistent between the cohorts when evaluated by Spearman's rank correlation analysis using all SNP-transcript pairs (Cor = 0.25, $p < 2 \times 10^{-16}$). To select a conservative set of candidate loci based on this property for further analysis, we identified the SNP-transcript pairs that were in the top 1% of the delta-$R_g^2$ distribution and also showed evidence of a strong *cis*-acting eQTL (genotype effect $p < 5 \times 10^{-8}$), in both cohorts. Data for the resulting set of 146 SNP-transcript pairs, including $R_g^2$ values, are provided in Table 2 via the Interactive Results Tool (which is also available to download from Zenodo and shown within *Supplementary file 1*).

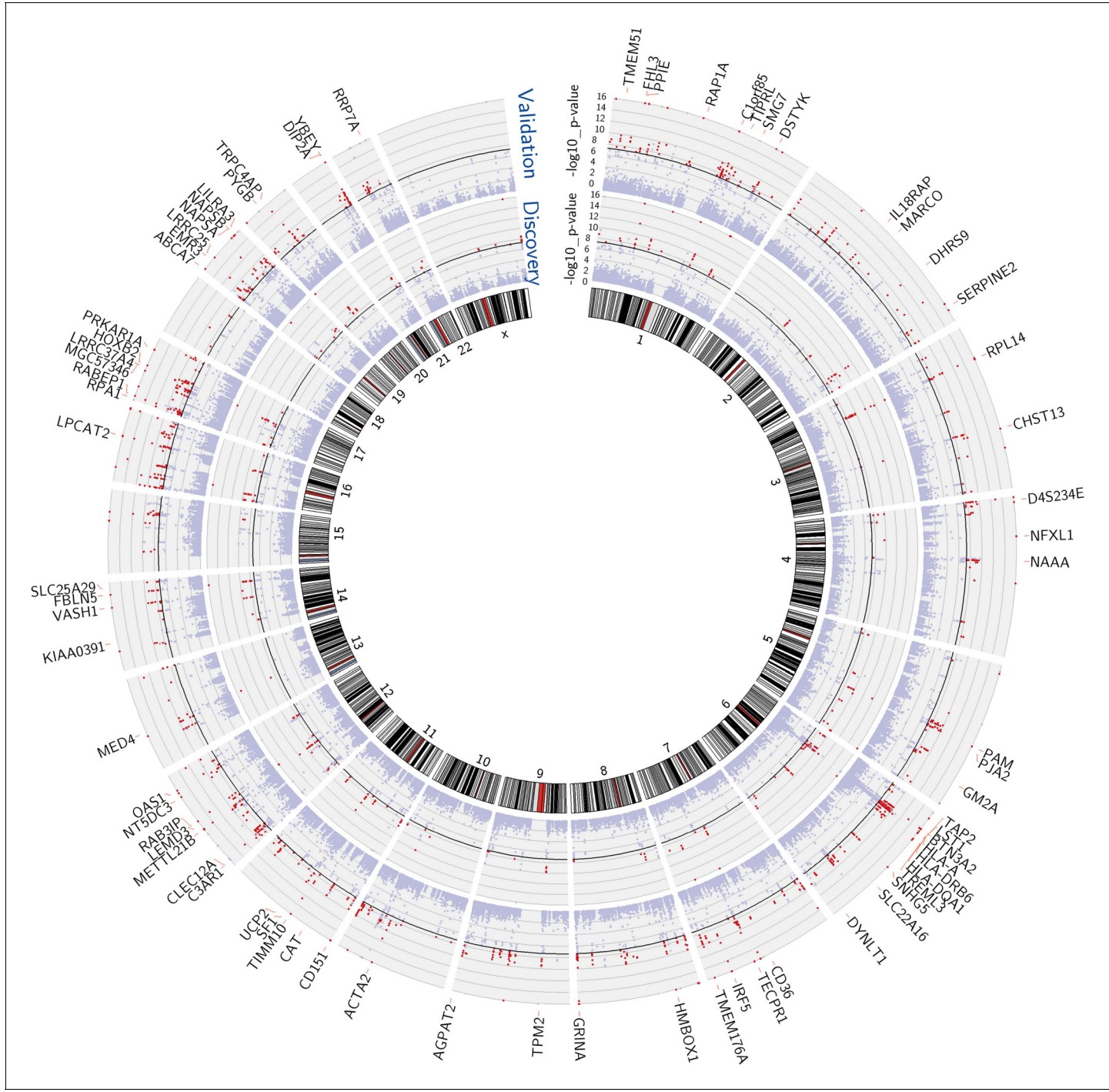

**Figure 1.** Multiple genes show both a transcriptional response to the vaccine and evidence of genetic regulation of gene expression (cis-acting eQTL) in both cohorts. Manhattan plots of the genotype-expression—log10 p-values across the genome for the discovery (inner circle) and validation (outer circle) cohorts. Each dot represents a SNP-transcript pair. Red dots indicate SNP-transcript pairs for which there is evidence of significant genotype-expression association (genotype $p < 5 \times 10^{-8}$) and evidence of a transcriptional response to the vaccine (day effect $p < 0.05$). The 78 genes that showed both properties in the two cohorts are shown in the outer margin.

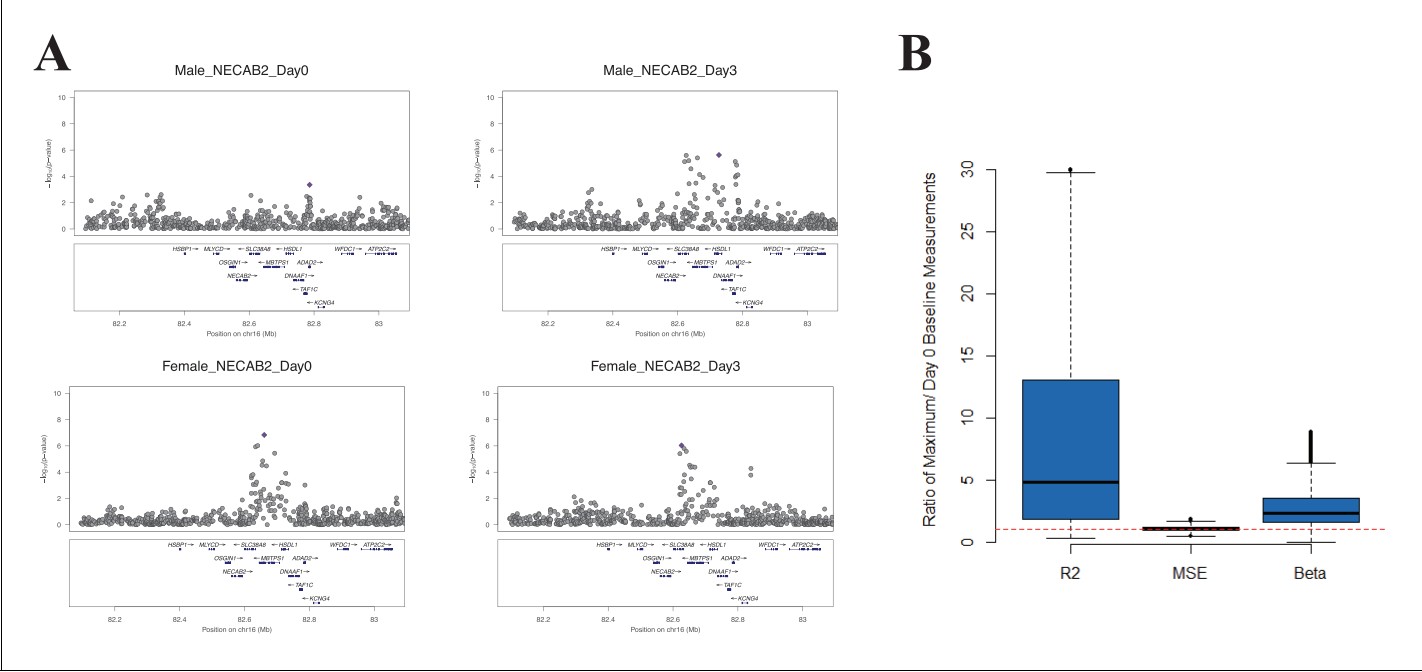

**Figure 2.** At some loci, the magnitude of the genetic effect changes after the experimental perturbation. (**A**) A specific example of this phenomenon: local Manhattan plots for the gene *NECAB2* before and on day 3 after vaccination in each of the two cohorts, showing an increase in the magnitude of the genotype effect ($R^2_g$) after the experimental perturbation. (**B**) An increase in $R^2_g$ after the experimental perturbation is a general feature of the SNP-transcript pairs that show a strong *cis*-eQTLs and a transcriptional response to vaccination (left). The within-genotype variance is unchanged (MSE, center), while the strength of the genotype effect on expression (slope of the additive association; β, right) increases, suggesting that the latter is the main driver for the observed increase in the genetic effect after vaccination.

## Content analysis shows enrichment for genes involved in membrane trafficking, antigen processing, and antigen presentation

Of the 78 genes that had the strongest validated evidence of a genotype effect and a transcriptional response to the vaccine, 14 were also in the list of 34 genes with the strongest evidence of an increase in the magnitude of the genetic effect after vaccination. Content analysis on the union of the two sets (98 genes) showed significant enrichment for genes involved in antigen processing and presentation, cytotoxic T-lymphocyte-mediated apoptosis of target cells, dendritic cell maturation and function, and membrane trafficking (*Figure 3*).

## Integration of genotype, expression, and antibody titer data identifies 20 genes with the strongest evidence for genetic variation influencing the humoral immune response to influenza vaccination

We and others have shown that for some transcripts there is significant correlation between the magnitude of the transcriptional and antibody responses to the vaccine stimulus (*Zhu et al., 2010*; *Bucasas et al., 2011*; *Nakaya et al., 2011*) In a combined analysis of the two cohorts in the present study, 301 transcripts were found to correlate with the magnitude of the antibody response (*Figure 4*). Additional details of these 301 transcripts, including correlation coefficients and days of maximum correlation, are provided in Table 3 via the Interactive Results Tool (which is also available to download from Zenodo and shown within *Supplementary file 1*). We imposed an additional selection threshold based on this correlation, and identified 20 genes that show evidence of significant genotype-expression association (genotype effect $p < 5 \times 10^{-8}$), a significant correlation between the transcriptional and antibody responses (expression-antibody effect $p < 0.05$), and either a transcriptional response to the vaccine (day effect $p < 0.01$) or evidence of a change in the magnitude of the genetic effect after vaccination (top 1% of the delta-$R_g^2$ distribution) in the two independent cohorts. These loci have the strongest evidence of genetic variation influencing the immune

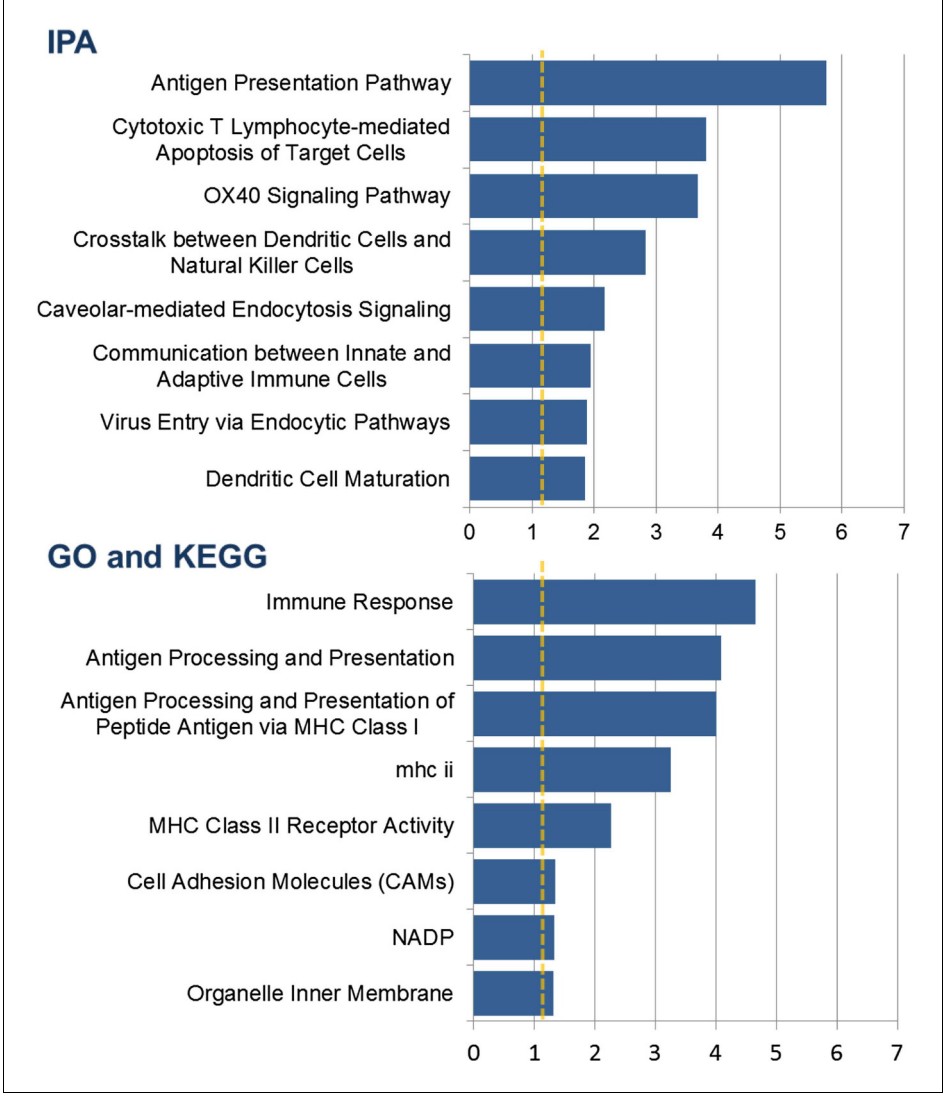

**Figure 3.** Content analysis shows enrichment for genes involved in membrane trafficking, antigen processing, and antigen presentation. Barplots show categories with significant overrepresentation in the list of 98 genes with a strong *cis*-eQTL and a response to vaccination expressed as either a transcriptional response or a change in the genetic effect in both cohorts. The negative log(p-value) is plotted on the x-axis.

response to the vaccine, and include *TAP2, SNX29, FGD2, NAPSA, NAPSB, GM2A, C1orf85, JUP, FBLN5, CHST13, DIP2A, PAM, D4S234E, C3AR1, HERC2, LST1, LRRC37A4, OAS1, RPL14*, and *DYNLT1*. Remarkably, seven of these encode proteins involved in intracellular antigen transport and processing (*Figure 5*).

We determined genetic associations to the antibody response using 137 eQTL SNPs from these 20 loci. The quantile-quantile plot from the association tests performed on these SNPs shows marked deviation from the empirical null distribution for QTL associations (*Figure 6*), supporting the idea that these loci are enriched for true genetic associations.

## The study design permits causal and reactive model analyses

We explored three types of associations in our work: genotype to gene expression (eQTL), gene expression to antibody titer, and genotype to antibody titer (QTL). We now considered alternative models for the relationships between these distinct types of association (*Figure 7A*), and we evaluated our data to determine which of these alternatives appears most consistent with our

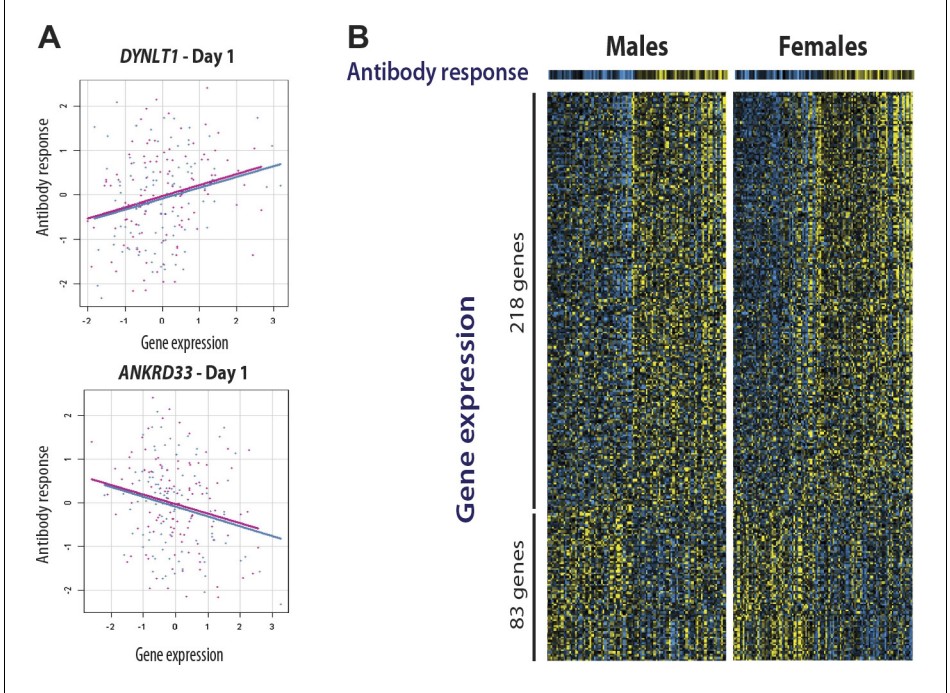

**Figure 4.** Gene expression at specific loci correlates with the antibody response to vaccination. (**A**) Examples of positive (*DYNLT1*) and negative (*ANKRD33*) correlation between gene expression on day 1 and the magnitude of the antibody response to the vaccine. Data points and regression lines in the scatterplots display the results for the discovery (blue) and validation (magenta) cohorts. (**B**) A total of 301 genes showed evidence of significant correlation between gene expression and the antibody response to the vaccine in both cohorts. Of these, 281 showed evidence of positive correlation and 83 of negative correlation. Each individual is represented by a column in the heatmaps. The top heatmaps display the magnitude of the antibody response (titer response index). The bottom heatmaps display the deviations around the expression mean for each gene. Individual gene identifiers and correlation coefficients are presented in the Interactive Results Tool.

observations. The alternative models considered were: (i) genotype association with gene expression is independent of genotype association or trends of association with antibody response (independent model); (ii) genotype association or trends of association with antibody response are mediated by gene expression patterns that are strongly correlated with genotype (causal model); and (iii) genotype associations to antibody response are not mediated by expression, but instead gene expression patterns are a response to the antibody trait or its early correlates (reactive model). To perform a comparative analysis of these alternatives we extended the framework for causal modeling (*Pearl, 2010*) in eQTL data recently developed by others (*Millstein et al., 2009*) and applied the method to our time-course gene expression study. We used the 137 eQTL SNP-transcript pairs from the 20 loci with the strongest evidence of genetic variation influencing the immune response to the vaccine, as described above. We found that the patterns in the data trend toward the causal model compared to the reactive model (*Figure 7B*), but a power analysis based on the distribution of the empirical effect sizes of our observed associations also indicates that our sample size is too modest to support definitive conclusions (*Figure 7C*).

## Discussion

The results provide an unbiased integrative survey of the genetic and transcriptional components of the humoral immune response to influenza vaccination in humans. They suggest that variation at the level of genes involved in antigen processing and intracellular trafficking is an important determinant of vaccine immunogenicity. Even in healthy, young individuals, there are a significant number of people who do not develop a protective antibody response after influenza vaccination. If these individuals could be identified prior to vaccination, modifications to the type or dose of vaccine could be attempted, with the goal of reducing the number of unprotected vaccinated individuals. The genes

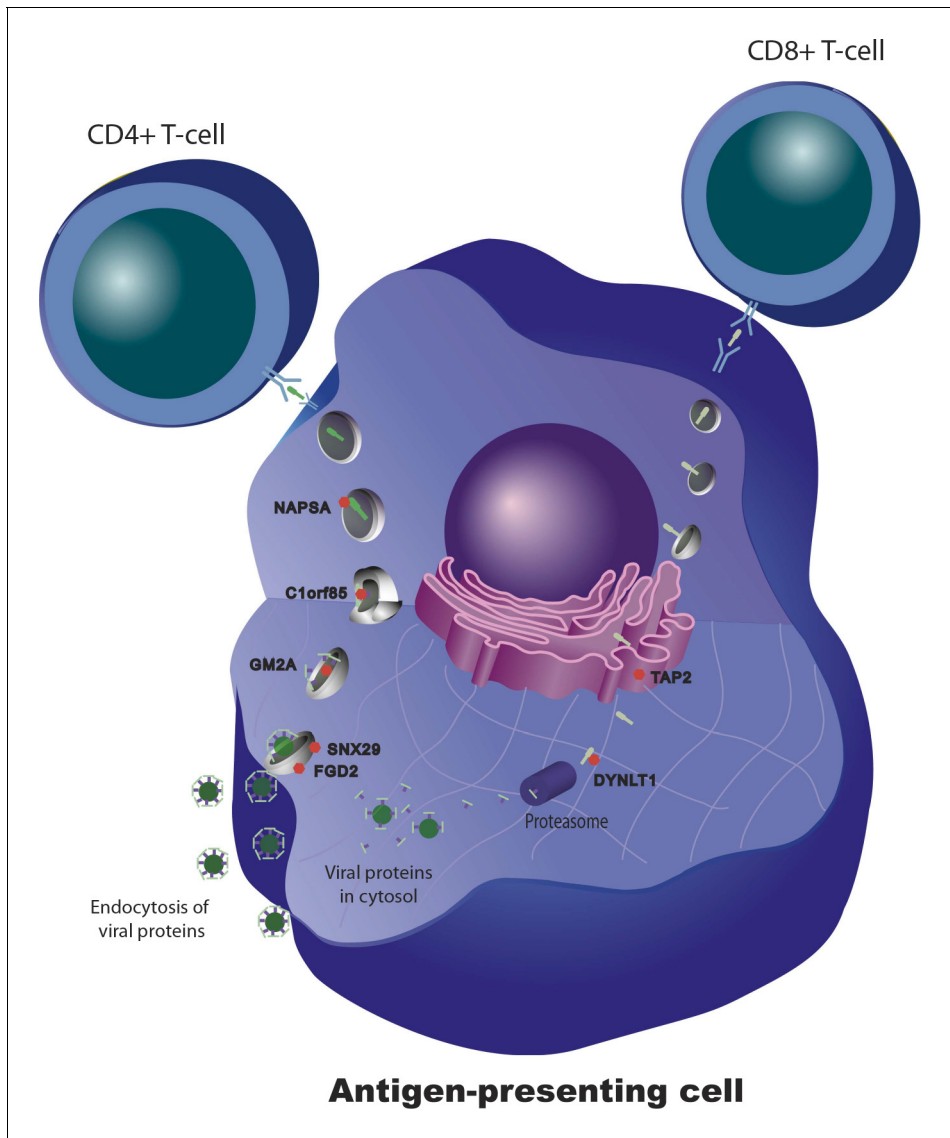

**Figure 5.** Genetic variation in intracellular antigen transport and processing influences the human immune response to influenza vaccination. 20 genes show evidence of a transcriptional response to vaccination, significant genotype effects on gene expression, and correlation between the transcriptional and antibody responses. Remarkably, seven of these are involved in intracellular antigen transport, antigen processing, and antigen presentation.

identified in this study as playing a role in variation in the humoral response to vaccination would be a logical starting point for the development of DNA- or RNA-based predictive biomarkers. Prospective evaluation of such biomarkers would be the next step towards clinical implementation.

Understanding the mechanisms that underlie variation in response to the vaccine may also direct modification of factors that enhance the response. Most of the efforts to date have focused on vaccine adjuvants that activate known immunologic mechanisms. Surprisingly, many of the genes identified in this study encode proteins that are not specifically immune but play a more general role in membrane trafficking and intracellular transport. Interventions aimed at increasing vaccine antigen affinity to these proteins or altering their intracellular concentrations could represent new avenues in vaccine development.

More broadly, the results demonstrate that a longitudinal, integrative genomic analysis study design, applied to a clinical intervention, is an efficient alternative to cross-sectional methods for the identification of genes involved in medically relevant complex traits. By making repeated measurements on the same individual over time after a controlled experimental perturbation, we were able

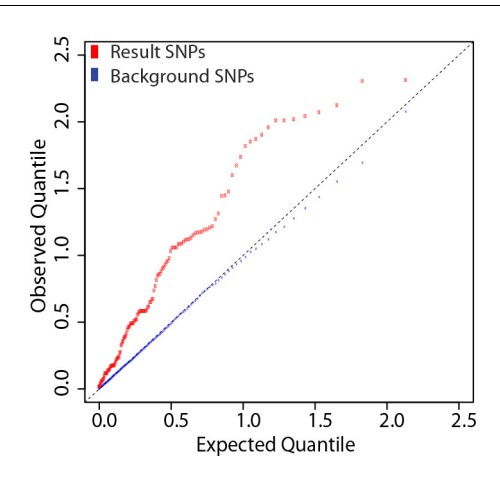

**Figure 6.** SNPs at the 20 loci identified show evidence of association with the antibody response to the vaccine. 137 SNP-transcript pairs with evidence of a strong *cis*-eQTL, a dynamic response to the vaccine (a change in transcript abundance or in the magnitude of the genetic effect), and correlation between the transcriptional and antibody responses were selected (result SNPs, in red). The empirical quantile-quantile plot of the result SNPs shows significant deviation from the empirical distribution of the entire data set (background SNPs, in blue).

to account for individual variation in a way that would not have been possible otherwise. The dynamic nature of the measurements also allowed us to uncover genetic effects that are either enhanced by or only evident after the experimental perturbation. The specificity of gene identification in this study emerges from the genome's acute response to the perturbation, which cannot be assessed by a cross-sectional eQTL analysis or a genome-wide association study. This approach could be used for a broad variety of medically important problems whenever there is the opportunity to test a well-controlled intervention such as drug, dietary, or vaccine responses.

Several limitations of the study are worth noting. First, we studied two samples of healthy young adults, thereby excluding the segments of the population that are most likely to have a poor response to influenza vaccination: children, the elderly, and individuals with severe illnesses. Second, in order to minimize the risk of false associations related to population stratification, we studied an ethnically homogeneous group of individuals. Third, while an interesting aspect of our study design is that it could open the door for direct comparisons of causal and reactive models, the sample size in this study was not sufficient to establish whether or not there is a causal relationship between the loci for which an association was identified and the antibody response to the vaccine. Finally, while antibody titers have historically been used to evaluate vaccine responsiveness, it is clear that they do not capture the complexity of the human immune response to vaccination. Additional studies would be necessary to determine whether the genes identified are also related to variation in influenza vaccine responses in groups other than the one chosen for this study, whether there is a causal relationship between these genes and the antibody response, or whether they also influence the cell-mediated immune response to the vaccine.

## Materials and methods

### Visual summary

A visual representation of the study design, the resulting data sets, and the integrative analysis scheme, is presented in *Figure 8*.

### Study subjects

Healthy volunteers ages 18 to 40 years were enrolled. Individuals who were known to have received an influenza vaccine in the previous 3 years or who had signs or symptoms of an active infection at the time of enrollment were excluded. To minimize false-positive results related to population stratification, enrollment was limited to individuals of self-reported Caucasian ancestry. Enrollment, vaccination and sample collections were conducted at a university campus. The initial (discovery) cohort was restricted to males. The validation cohort was enrolled approximately 18 months after the initial cohort and was restricted to females. The protocol was approved by the institutional review boards of all participating institutions. Informed consent was obtained from each subject prior to enrollment.

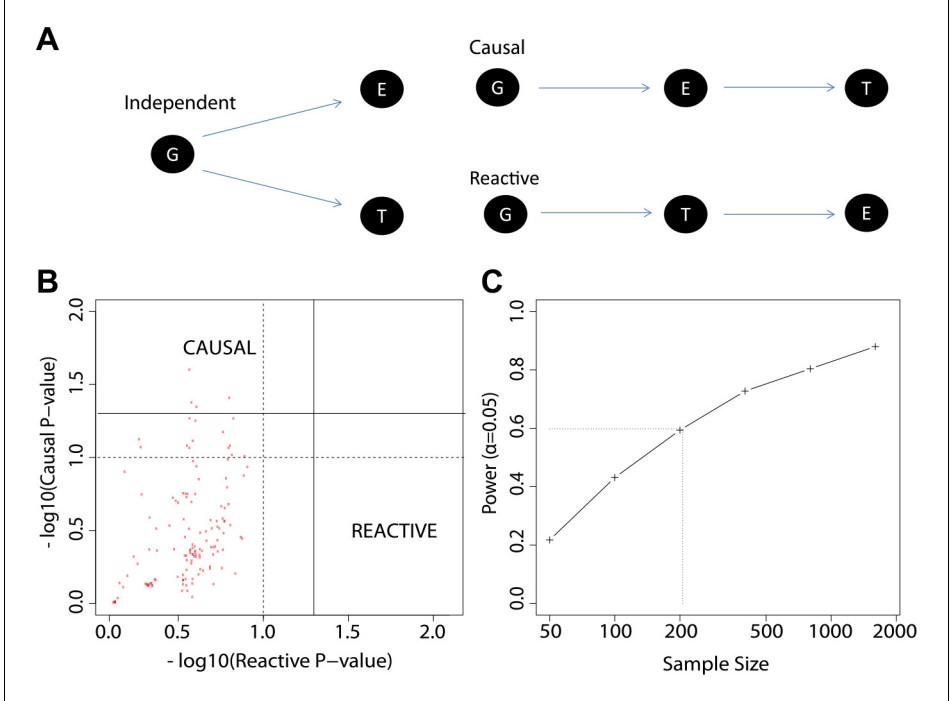

**Figure 7.** The study design permits causal and reactive model analyses. (**A**) Three models were evaluated, each showing a candidate hypothesis for the three-way association between genotype (G), expression (E) and trait (T). In the independent model, expression and trait each associate with genotype but are not themselves directly related. In the causal model, expression mediates the association between genotype and trait. In the reactive model, genotype and expression relate through the trait, so that gene expression changes are a downstream response to the trait. (**B**) p-values for independent-versus-reactive and independent-versus-causal hypothesis tests. Each point shows the result for one SNP-transcript pair. Points to the right of the solid vertical line are significant (p<0.05) for the reactive hypothesis and points above the solid horizontal line are significant for the causal hypothesis. The dashed line shows a p=0.1 threshold. (**C**) Power for rejection of the independent hypothesis. Non-independent data were simulated with effect sizes and variances similar to those in the enrichment set (the set of SNP-transcript pairs that were found to be significant in our study). The curve shows the proportion of cases in which the simulated data rejected the independent (null) hypothesis. The dotted line indicates the combined sample size in our study.

## Vaccine

Study participants were immunized on day 0. Those enrolled in the initial cohort received the 2008–2009 inactivated trivalent influenza vaccine (A/Brisbane/59/2007[H1N1], A/Brisbane/10/2007[H3N2], B/Florida/4/2006; Sanofi-Pasteur, Lyon, France). The validation cohort received the 2009–2010 vaccine, which came from the same manufacturer and included (A/Brisbane/59/2007), (A/Brisbane/10/2007[H3N2]), and (B/Brisbane/60/2008) strains.

## DNA samples

Whole blood samples (7 ml) for DNA purification were collected in Vacutainer acid citrate dextrose (ACD) tubes (Beckton-Dickinson, Franklin Lakes, NJ), on day 1 after vaccination. DNA was purified using Qiagen Gentra Puregene Blood Kits (Qiagen Sciences, Germantown, MD). Quantitation and quality control were performed with a NanoDrop-1000 spectrophotometer (Thermo Fisher Scientific, Waltham, MA) and using the Quant-iT PicoGreen dsDNA Reagent (Life Technologies, Carlsbad, CA) in a Tecan GENios microplate reader (Tecan Group, Mannendorf, Switzerland).

## RNA samples

Peripheral blood samples for RNA purification were obtained immediately before (day 0) and on days 1, 3, and 14 after vaccination. To minimize changes in gene expression induced by sample handling and processing, whole-blood samples (2.5 ml) were collected in PAXgene RNA stabilization tubes (Qiagen Inc., Valencia, CA) and frozen at −80°C. RNA purification was performed using the

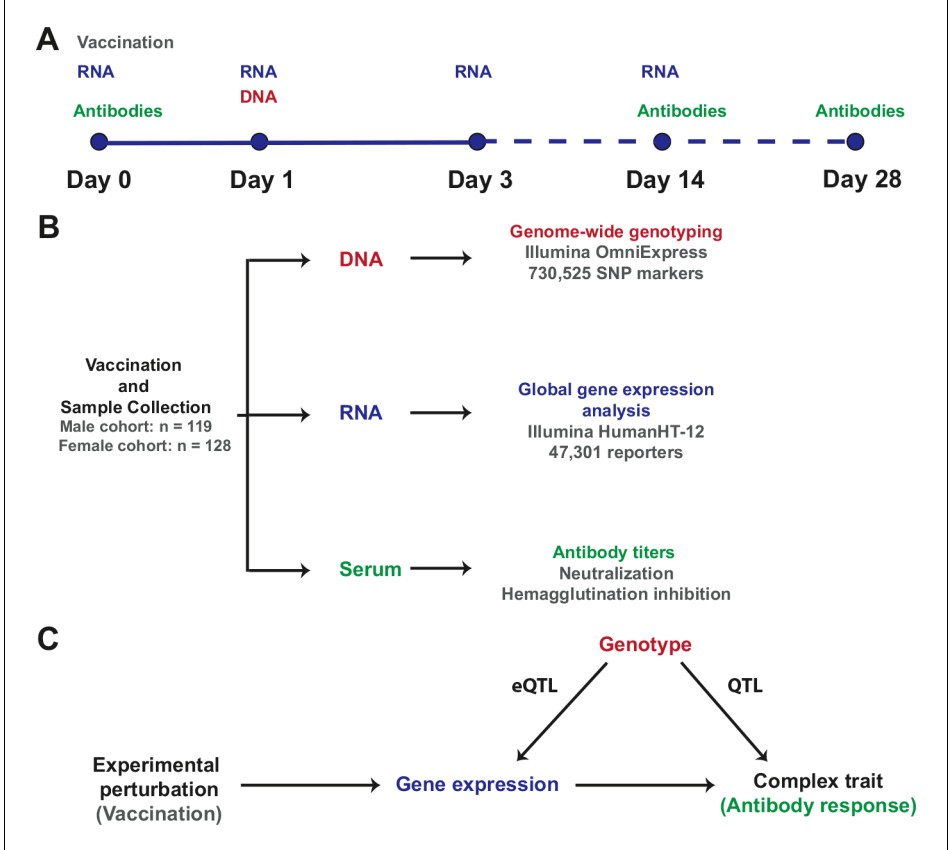

**Figure 8.** Study design and integrative analysis scheme. (**A**) Individuals were immunized on day 0 and peripheral blood RNA samples were obtained on days 0, 1, 3, and 14. Antibody titers were measured on pre-immune sera and on days 14 and 28. Genotyping was carried out on a peripheral blood genomic DNA sample obtained on day 1. Identical sample collection schemes were used, 1 year apart, for the discovery (males) and validation (females) cohorts. (**B**) Sample sizes and data generation platforms. (**C**) Integrative analysis involved identification of loci that exhibit a transcriptional response to vaccination, evidence of genetic regulation of expression (constitutive eQTL), evidence of correlation between gene expression and the antibody response, and evidence of correlation between genotype and the antibody response (QTL). Because transcript abundance was measured serially, we were able to evaluate changes in the magnitude of the genetic effect on expression at different time points following vaccination. In addition, the study design permitted QTL analysis conditional on gene expression, which led to the identification of loci whose genetic effects on the antibody response are causally linked through the eQTL.

PAXgene Blood RNA system (Qiagen Inc.). All RNA samples from an individual were consistently purified in the same batch. Spectrophotometry (NanoDrop-1000 Spectrophotometer; Thermo Fisher Scientific) and microfluidic electrophoresis (Experion Automated Electrophoresis System; Bio-Rad Laboratories, Hercules, CA) were used for quality control.

## Whole-genome genotyping

Genotyping was performed on Illumina HumanOmniExpress microarrays (Illumina, Inc., San Diego, CA) following the manufacturer's instructions. The arrays include 730,525 SNP markers. Basic quality control of the genotyping data was performed on GenomeStudio software, version 2010 (Illumina, Inc.). All microarrays had call rates >0.99.

## Measurement of global transcript abundance

In vitro transcription was performed using Ambion Illumina TotalPrep RNA Amplification Kits (Applied Biosystems/Ambion, Austin, TX). cRNA was hybridized onto Illumina HumanHT-12v3 or HumanHT-12v4 Expression BeadChips (Illumina, Inc.), following the manufacturer's protocol. All samples for a given individual were processed on the same slide. The arrays have 48,742 (v3) and 47,301 (v4) reporters, representing approximately 25,000 genes and non-annotated gene candidates.

## Serum samples and antibody titer measurements

Whole blood (10 ml) was collected in Vacutainer Serum Separator Tubes (Beckton–Dickinson). Serum was separated by centrifugation prior to storage at −20°C. Hemagglutination inhibition (HAI) tests were performed as previously described (*Dowdle et al., 1979*), except for a starting serum dilution of 1:4 and the use of turkey red blood cells. HAI test antigens were allantoic fluid harvests from infected embryonated hen's eggs (whole-virus antigens). Neutralizing antibody tests were performed as previously described (*Frank et al., 1980*) except that hamster serum was not included. Test strains were the same as those used in the vaccine.

## Antibody titer data analysis

HAI and neutralizing antibody titers were measured for each viral antigen included in the vaccine. For all antigens, the change in antibody titer post-vaccination is known to be negatively correlated with the pre-vaccination titer. The responses to the individual viral antigens were correlated within individuals. For each individual, we computed a Titer Response Index (TRI), as previously described (*Bucasas et al., 2011*). The TRI characterizes the magnitude of an individual's antibody rise accounting for their pre-vaccination titer and integrating the responses across the three vaccine components to provide an overall measure of vaccine responsiveness.

## Expression data processing and quality control

Raw signal intensity data from all individuals and all time points in the discovery cohort were first processed in a single batch. Background adjustment, variance stabilization transformation (*Lin et al., 2008*) and robust spline normalization were performed using the *lumi* package (*Du et al., 2008*) in R (*R Development Core Team, 2009*). Eight individuals had missing expression data from two or more time points and were excluded. We required a detection p-value of <0.01 in at least 80% of the samples for a transcript to be considered detected. We also aligned the entire set of expression reporter sequences to the human genome reference sequence (Build 36 [March 2006]/hg18) by applying the BLAT algorithm in BlatSuite34 software (*Kent, 2002*), and excluded any reporters that did not map or mapped to more than one region. Using these two thresholds for the data in the discovery cohort, the final data set included 9809 detected transcripts. This data set was then used for eQTL analysis in the discovery cohort.

   Once data generation for the validation cohort was completed, the expression microarray signal intensity data from all individuals and all time points in that cohort were processed in a single batch. Background adjustment, variance stabilization transformation, robust spline normalization, and the application of detection thresholds were performed identically to the discovery cohort. Five individuals had missing expression data from two or more time points and were excluded. Because two different array versions were used (HT12-v3 and HT12-v4), unique reporter identifiers (ProbeID and nuID) for the 9809 reporters selected in the discovery cohort were used to subset the data from the validation cohort. This data set was then used for eQTL analysis in the validation cohort.

   The analysis of correlation between gene expression and antibody titer in the discovery cohort was previously published (*Bucasas et al., 2011*). As part of the integrative genomic analysis described in the present study, we performed a similar analysis of this expression/titer correlation, but included expression data from both cohorts. For this purpose, the two data sets described above were combined, and an additional quantile normalization step was performed to account for batch effects between cohorts.

## Genotyping data processing and quality control

Array quality was initially assessed using GenomeStudio software (Illumina, Inc.). Default algorithms were used to normalize, generate SNP clusters, and make genotype calls. SNPs with minor allele frequency (MAF) <0.05 and Hardy-Weinberg Equilibrium (HWE) $\chi^2 < 1 \times 10^{-7}$ were removed prior to analysis. For local eQTL mapping, we restricted the analyses to SNPs within 1 Mb (500 kb upstream or downstream) of each unambiguously mapped expression reporter sequence. We used the genotype information to reliably establish that the individuals in each cohort were indeed unrelated and ethnically matched. For this, we estimated pairwise identity-by-descent (IBD) metrics on the basis of identity-by-state (IBS) information from their genome-wide SNP genotyping results, followed by multidimensional scaling (MDS) analysis on the resulting matrix of pairwise distances, using PLINK

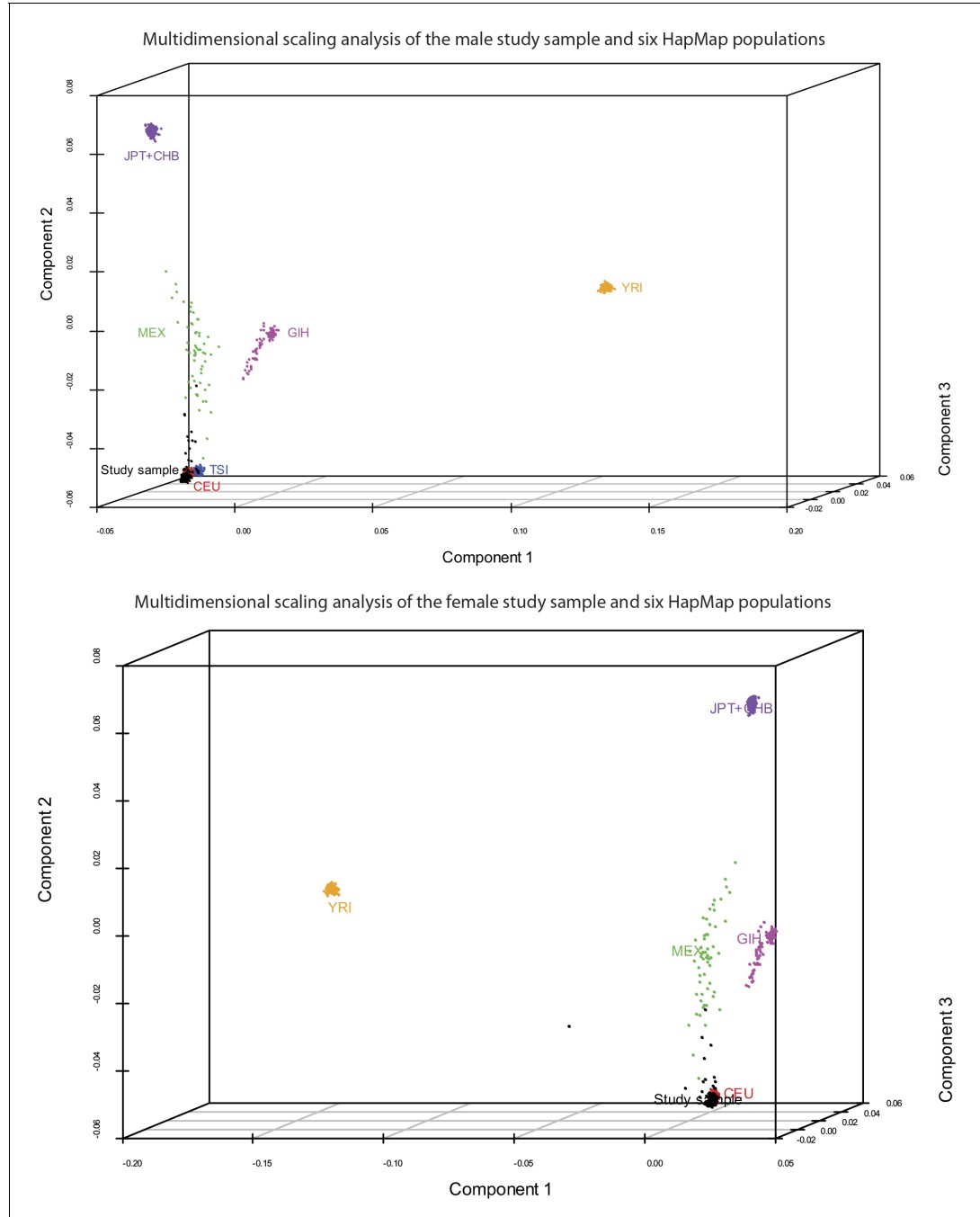

**Figure 9.** Study samples cluster with the HapMap CEU population. Pairwise identity-by-descent metrics were estimated based on genotype data from our two study samples and six HapMap populations. Multidimensional scaling analysis was performed on the resulting pairwise distances. Components 1–3 of this analysis are plotted for the male (top) and female (bottom) cohorts, and the comparison populations. As expected, the study samples cluster with the HapMap CEU population. 12 outliers were identified in each cohort and were removed prior to analysis.

(*Purcell et al., 2007*). The first three components of the MDS analysis on our study samples and the founders from six HapMap populations were plotted with R package Scatterplot3D (*Ligges and Machler, 2003*) and are shown in *Figure 9*. As expected, our study samples from both cohorts cluster with the HapMap CEU population. In the male cohort, we identified four pairs of individuals with cryptic or undisclosed familial relationships (pi-hat ≥ 0.125). 12 outliers were identified by component 2; two of the four pairs of cryptic or undisclosed relatives were among the outliers. One

individual from each related pair and all remaining outliers were removed prior to analysis. In the female cohort, we identified one pair of individuals with cryptic or undisclosed familial relationship (pi-hat $\geq$ 0.125). 11 outliers were identified by component 2 and one by component 1. One individual from the related pair and all outliers were removed prior to analysis.

## Integrative genomic analysis

Analyses were carried out separately for each cohort and in the combined data set. Because we have repeated gene expression measurements on the same individuals over a series of time points, we performed a random effects linear model analysis. For each SNP-transcript pair, we fit a model with terms for day, additive effects for genotype and day–genotype interactions, and a random effect for each person:Where $Y$ = observed expression value, $i$ = day (0, 1, 3, 14), $j$ = individual, $D$ = day effect, $g$ = genotype effect, $P$ = person effect (random), and $e$ = random error. We fit this model using the R package nlme and the estimation method of the same name. The day effect ($D$) is a shift term which considers time-ordered change in expression. The genotype effect ($g$) is a main effect for genotype that considers the expression differences between genotypes to be additively related to the number of copies of a reference allele. The day–genotype interaction allows the additive effect to vary by day (see *Figure 10* for a visual representation of the analysis scheme). The person effect ($P$) is modeled as a random effect; this term helps to efficiently account for the fact that the same individual appears in each time point, without overfitting the expression model to the particular sample ascertained in our study. We explored different parametrizations, such as allowing the person effect to vary by day, and found that the simplest model with a single term for each person was effective. This person effect can account for inter-individual or batch differences which are stable within individuals for the duration of the study.

$$Y_{i,j} = D_i + \beta g_j + \beta_i g_j + P_j + e_{i,j}$$

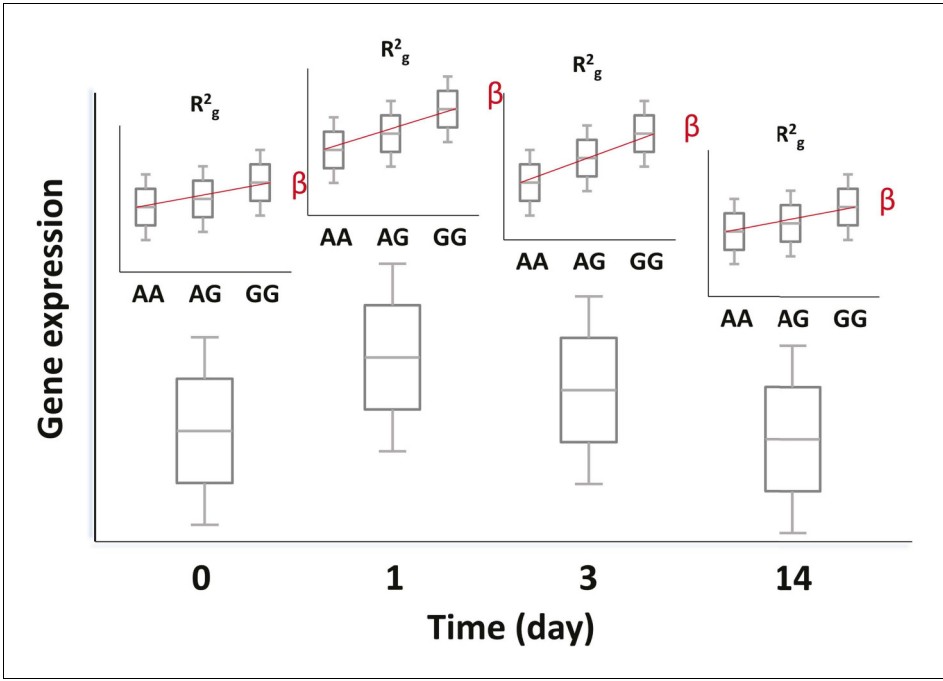

**Figure 10.** Genetic and transcriptional analysis on a prospective cohort. The figure displays hypothetical results for a single SNP-transcript pair. For any such pair, one may observe changes in transcript abundance at different time points after the experimental perturbation (lower box plots). In addition, gene expression at each time point may be different for different genotypes when there is evidence of an eQTL (upper box plots). Finally, the slope of the expression-genotype association (β), as well as the proportion of the variance in expression explained by genotype ($R^2_g$), may vary across time points.

In addition to the random effects linear model, we analyzed each expression trait with respect to genotype at each time point using where $Y_j$ denotes the matrix of expression traits of individuals $j$ at a given time point (days 0, 1, 3, 14) and $G_{i,k}$ is a matrix of genotypes for individuals $j$ at SNP locus $k$ such that each element is assigned 0, 1 or 2 according to the number of minor alleles at the $k$th locus of the $j$th individual. This allowed estimates of the proportion of expression variance explained by individual genotypes from the model's coefficient of determination ($R^2_g$). This model permitted a more detailed examination of the changes in the strength of the genotypic association with expression at each time point. To detect SNP-transcript pairs where the magnitude of the genotype effect varied after immunization, we took the difference in $R^2_g$ measures ($\Delta R^2_g$) from pairwise linear models fitted after (days 1, 3 and 14) and before vaccination (day 0). We then took the top 1% $\Delta R^2_g$ values between day 0 and a later time point. This *cis*-acting eQTL subset represents the loci that show significant changes in the genetic effect as a result of the vaccine.

$$Y_j = \beta G_{i,k} + e$$

## Correlation of transcripts with the humoral immune response

The titer response index was treated as a dependent variable and modeled as a linear combination of expression values for each transcript across the time course. A single F-statistic p-value was determined to evaluate the explanatory value of the expression data using all days. Separate evaluations were made in each cohort and in the combined data. The 301 genes displayed in *Figure 4B* had F-statistic p-values in the combined data <0.01 and absolute values of the maximum average correlation between expression and titer response in the two cohorts >0.15.

## Content analyses

Enriched biological and functional pathways were analyzed using DAVID Bioinformatics Resources (*Dennis et al., 2003*), and Ingenuity Pathway Analysis (IPA) software.

## Causal and reactive model analyses

To evaluate the relationship between the associations identified in our study, we extended a recently published analysis framework for causal modeling in eQTL data (*Millstein et al., 2009*). The relationship is modeled as where $T$ is trait, $E$ is gene expression, and $G$ is genotype at a locus. In our experiment, the term $E$ includes a separate value for each day. The term $G$ has a separate effect on the trait under the independent null model, while under the causal alternative, where it acts through gene expression, its influence is entirely transmitted through $E$. The statistic used to assess the influence of $G$ is the partial F reported by the 'aov' function in R (version 2.15.1). The analyses utilize an equivalence test approach, where the null hypothesis is that the second independent variable has an effect on the response conditional on the first variable, while the alternative is complete co-linearity in the associations. The null distribution of the F statistic–which under parametric assumptions would have a non-central F distribution—is derived using a permutation test procedure, as follows: The relationship between $E$ and $T$ is decoupled by regressing $E$ on $G$, permuting the resulting residuals, then adding the permuted residuals to the predicted values to arrive at $E^*$, which is independent of $T$ but maintains the marginal variance and $G$-correlation of $E$. This procedure enforces the assumption of the independent model, while leaving other properties unchanged. The partial F statistic for $G$ in represents a sample from the distribution under the independent null hypothesis. This process, repeated 1000 times, provided a null distribution. The p-value for the observed F is the proportion of the null distribution that is below the observed value.

$$T \sim E + G$$

$$T \sim E^* + G$$

The reactive model is evaluated using the same approach, but with the expression and trait terms swapped, that is

$$E \sim T + G$$

This scheme tests the extent to which the trait value mediates the relationship between genotype and gene expression. In this analysis, the permutation step decouples the expression-to-trait

relationship from the eQTL association and asks if the eQTL association disappears after accounting for the expression-to-trait association.

Power analyses for these models apply the same scheme to simulated data. The term $G$ is simulated as binomial, while $E$ and $T$ are normal, with parameters chosen to match the minor allele frequency, variance, and correlations observed in our data set. The power to reject the false (independent) model is shown using a threshold of $p < 0.05$. The power properties for the tests used to evaluate both the reactive and causal models are equivalent because the model structures and $r^2$ relationships are symmetric between the two alternatives.

## Acknowledgements

The clinical trials were conducted at the Beutel Health Center at Texas A&M University with the strong support of Dr Martha Dannenbaum, Scott Draper, and Richard Darnell. Mr Uri Kelman of Kelman Design contributed to the generation of *Figure 5*.

## Additional information

### Funding

| Funder | Grant reference number | Author |
| --- | --- | --- |
| US National Institutes of Health - National Institute of Allergy and Infectious Diseases | NO1-AI-030039 | Robert B Couch John W Belmont |
| US National Institutes of Health - National Institute of Allergy and Infectious Diseases | 1K23AI087821-01 | Luis M Franco |
| US National Institutes of Health - Ruth L Kirschstein National Research Service Award | T32 GM07526 | Luis M Franco |
| US National Institutes of Health - Ruth L Kirschstein National Research Service Award | F31AI071372-03 | Kristine L Bucasas |

The funders had no role in study design, data collection and interpretation, or the decision to submit the work for publication.

### Author contributions

LMF, Acquisition of data, Analysis and interpretation of data, Drafting or revising the article; KLB, Acquisition of data, Analysis and interpretation of data, Drafting or revising the article; JMW, Acquisition of data; DN, Acquisition of data; XW, Acquisition of data; GEZ, Acquisition of data; NA, Acquisition of data; JMQ, Acquisition of data; AR, Analysis and interpretation of data; PY, Analysis and interpretation of data; MSB, Acquisition of data, Drafting or revising the article; RBC, Conception and design, Acquisition of data, Analysis and interpretation of data, Drafting or revising the article; JWB, Conception and design, Acquisition of data, Analysis and interpretation of data, Drafting or revising the article; CAS, Analysis and interpretation of data, Drafting or revising the article

### Ethics

Human subjects: The protocol was approved by the institutional review boards of all participating institutions. BCM IRB identifiers: H-28980, H-25352. Informed consent and consent to publish was obtained from each subject prior to enrollment.

## Additional files

### Supplementary files

• Supplementary file 1. Compilation of the tables and figures (XLS). This is a static version of the Interactive Results Tool, which is also available to download from Zenodo (see major datasets).

## Major datasets

The following datasets were generated:

| Author(s) | Year | Dataset title | Dataset URL | Database, license, and accessibility information |
|---|---|---|---|---|
| Franco LM, Bucasas KL, Wells JM, Niño D, Wang X, Zapata GE, et al. | 2013 | Time series of global gene expression after trivalent influenza vaccination in humansGEO SuperSeries GSE48024; | http://www.ncbi.nlm.nih.gov/geo/query/acc.cgi?acc=GSE48024 | Publicly available at GEO (http://www.ncbi.nlm.nih.gov/geo/). |
| Franco LM, Shaw CA, Belmont J | 2013 | Time series of global gene expression after trivalent influenza vaccination in humans (male cohort) GSE48018 (SubSeries of GSE48024); | http://www.ncbi.nlm.nih.gov/geo/query/acc.cgi?acc=GSE48018 | Publicly available at GEO (http://www.ncbi.nlm.nih.gov/geo). |
| Franco LM, Shaw CA, Belmont J | 2013 | Time series of global gene expression after trivalent influenza vaccination in humans (female cohort)GSE48023 (SubSeries of GSE48024); | http://www.ncbi.nlm.nih.gov/geo/query/acc.cgi?acc=GSE48023 | Publicly available at GEO (http://www.ncbi.nlm.nih.gov/geo). |
| Franco LM, Bucasas KL, Wells JM, Niño D, Wang X, Zapata GE, et al. | 2013 | "Interactive" version of data associated with the eLife paper "Integrative genomic analysis of the human immune response to influenza vaccination"10.5281/zenodo.6960; | https://zenodo.org/record/6960?ln=en#.UdPj-W1OItzY | Publicly available at Zenodo (https://zenodo.org/). |
| Franco LM, Bucasas KL, Wells JM, Niño D, Wang X, Zapata GE, et al. | 2013 | Adult influenza vaccine geneticsphs000635.v1.p1; | http://www.ncbi.nlm.nih.gov/projects/gap/cgi-bin/study.cgi?study_id=phs000635.v1.p1 | Publicly available at dbGAP (http://www.ncbi.nlm.nih.gov/gap). |

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
