## [Decision Letter]

Thank you for choosing to send your work entitled “Integrative genomic analysis of the human immune response to influenza vaccination” for consideration at *eLife*. Your article has been evaluated by a Senior editor and 2 reviewers, one of whom, Manolis Dermitzakis, is a member of our Board of Reviewing Editors.

The Reviewing editor and the other reviewer discussed their comments before we reached this decision, and the Reviewing editor has assembled the following comments based on the reviewers' reports.

The manuscript describes a project that aims to integrate genetic, gene expression, and vaccination data, together with immunological traits. The design is innovative and promises interesting discoveries, some of which are described in the submission. Some key concerns were raised that need to be addressed before a final decision can be made:

1) The eQTL discovery was performed in a cohort comprised of males only and the replication was performed in a cohort comprised of females only. What was the purpose of this design? The authors do not address the issue that discovery and replication are confounded by sex.

2) We suspect that an interaction term in the models would actually be more efficient in detecting genetic effects modified by the time point after vaccination. Why was this not used (instead a simple comparison among time points was made)?

3) The fact that there is enrichment in the QQ plots on Figure 6 does not provide evidence about causality of SNPs to vaccine response but simply of correlation among the two effects. More complex models need to be implemented to show causality.

4) The study design used two different array versions: Illumina HT-12v3 and HT-12v4. The authors should describe how they combined data from these arrays, particularly for transcripts where the probe sequence differs between versions.

5) The expression data processing should be described in more detail to make it clear how normalization, and so on, was performed given the different time points, array versions, and discovery/replication cohorts.

6) The authors mention that the person effect within their model would account for differences in cell populations among individuals, given that these are stable within individuals during the sampling timeframe. No data or references are provided to support this claim. Immune cell relative proportions can change rapidly.

7) The authors use the delta r^2^ metric to identify eQTLs that differ between time points or in response to the vaccination. It seems that the authors could have performed the analysis using the delta of transcription(time point x)−transcription(baseline) as a trait. This normalizes to each individual's starting point. Furthermore, a difference of r^2^ between time points can have multiple causes (as the authors point out). It seems to us that what they are most concerned with is cases where the effect size (beta) of a given eQTL differs between time points, *or* where the actual response (transcription time point x−transcription baseline) differs between individuals. The authors reasoning behind these choices should be made clearer.

8) For the trans-eQTL analysis, the authors consider as genome-wide significant any SNP-transcript pair with p<5×10^-8^. Indeed for a GWAS of common variation, this may be appropriate for a single trait, but the authors should account for testing nearly 10K transcripts.

[After resubmission, the editors also asked for the following comments to be addressed prior to acceptance.]

A) The revision is not satisfactory for point 7, above, on the delta r^2^. The case is still not convincing because only 146 of the 541 cases were validated. It is not clear why the difference in r^2^ is necessarily interpreted as a significant change when only 30% of the cases are validated.

B) The modelling as previously discussed in point 3 is somewhat weak and causality is not as well demonstrated, as the authors want to claim. The fact that the association with the residuals is lower than with the original values is not direct but indirect evidence that there are casual relationships. Also, how were the “random” SNPs chosen? Establishing a relevant null is important here. Also, direct causal models would help in establishing the relationships rather than the indirect approach the authors use.

---

## [Author Response]

*1) The eQTL discovery was performed in a cohort comprised of males only and the replication was performed in a cohort comprised of females only. What was the purpose of this design? The authors do not address the issue that discovery and replication are confounded by sex*.

Many loci show sex-specific effects at the level of gene expression. The study design for the discovery cohort, therefore, sought to minimize the effects of suspected variables, such as previous vaccine exposure, ethnicity, age, and sex. The follow-up study was designed to fill the resulting gap in information about responses to the seasonal influenza vaccine in women. We agree that the consequence of this feature of the data is confounding of potential sex-specific effects with other technical variables in the trials. However, it should not result in confounding with other biologically important variation, particularly in the validation of eQTLs. The literature is inconsistent with respect to terminology, but we prefer to make the distinction between the terms “validation” and “replication” (Igl et. al Hum Hered 2009; 67: 66-68).

Validation involves a confirmation attempt on a sample from a different population, where systematic variability (in this case, the gender difference that was noted by the reviewers) would be expected in addition to random variability. If a genetic association can be validated, it shows greater generalizability than a replicated association, in which the confirmation sample is drawn from the same population as the discovery sample. We agree that future studies should include both men and women as a way to identify potential sex-specific differences in transcriptional response to vaccine. Additional trials will also be required to assess the effects of age, other risk-increasing clinical variables, such as compromised immunity, and ethnic/racial origins.

*2) We suspect that an interaction term in the models would actually be more efficient in detecting genetic effects modified by the time point after vaccination. Why was this not used (instead a simple comparison among time points was made)*?

We agree with the reviewers and did attempt to investigate this. The model we employed for the analysis is as follows:Yi,j=Di+βgj+βigj+Pj+ei,j

This model includes interaction terms (β_i_*g*_j_) for the change in the coefficient for the additive effect of genotype by day. In the loci showing significant transcriptional response to the vaccine, we found that the main effect of genotype was far stronger than these interaction effects. The sign of the additive effect does not change across the days, but the magnitude tends to increase after vaccination. We therefore speculate that our sample size does not provide sufficient power to exploit the interaction terms. Future larger studies may allow more precise estimates of the time × gene interaction and allow identification of additional vaccine response loci beyond those detected here.

*3) The fact that there is enrichment in the QQ plots on Figure 6 does not provide evidence about causality of SNPs to vaccine response but simply of correlation among the two effects. More complex models need to be implemented to show causality*.

We agree with the reviewers that the enrichment shown in the QQ plots does not provide evidence of causality. As noted, the results do support the possibility that the selected markers are enriched for true associations to the humoral immune response to influenza vaccine. We have revised the manuscript to clarify this point and have separated Figure 6 from Figure 6 to help reinforce the difference in interpretation.

A potential causal relationship between the genes identified and the antibody response is supported by the conditional independence analysis shown in Figure 7 (formerly Figure 6). For this analysis, we first computed the R^2^ between SNP (genotype at SNPs that were validated eQTL and where gene expression was associated with the magnitude of the antibody response) and antibody response using an additive model. We then performed a regression analysis predicting antibody response from gene expression and we used the residuals from that analysis to perform another association analysis. We observed that the association between SNP and antibody, as measured by non-zero R^2^ values, disappeared after accounting for gene expression. This reduction in R^2^ was not observed in randomly-sampled sets of SNPs. The preceding data showed that the SNPs used are eQTL and that gene expression changes (e.g., day 1 post-vaccine) precede the change in antibody titers (day 28). Taken together, these observations support the interpretation that the predictive information in the genotype for the change in antibody titer is mediated by gene expression. This analysis supports a causal role for these genes in the change in antibody titer.

*4) The study design used two different array versions: Illumina HT-12v3 and HT-12v4. The authors should describe how they combined data from these arrays, particularly for transcripts where the probe sequence differs between versions*.

The sub-heading “Expression data processing and quality control” in the Materials and methods section has been re-written to more clearly explain how the expression microarray data were processed. In summary, for the initial part of the study (the eQTL component), the two cohorts were analyzed independently. Because of the difference in array versions, unique reporter identifiers (Illumina ProbeIDs and sequence-based nuIDs) were used to ensure that the same set of reporters was being analyzed in each of the cohorts. The two resulting expression data sets, which included data from the same set of 9,809 reporters, were then combined for the integrative genomic analysis component of the study.

*5) The expression data processing should be described in more detail to make it clear how normalization, and so on, was performed given the different time points, array versions, and discovery/replication cohorts*.

The sub-heading “Expression data processing and quality control” in the Materials and methods section has been re-written to more clearly explain how the expression microarray data were processed. The data for all individuals and all time points for a particular cohort were processed together. In the eQTL components of the analysis, the two cohorts were analyzed independently. Background adjustment, variance stabilization transformation, and normalization (VST-RSN) were carried out using identical parameters for each cohort. When the two data sets were combined for the integrative genomic analysis component, an additional quantile normalization was performed to adjust for batch effects between cohorts.

*6) The authors mention that the person effect within their model would account for differences in cell populations among individuals, given that these are stable within individuals during the sampling timeframe. No data or references are provided to support this claim. Immune cell relative proportions can change rapidly*.

We agree with the reviewers. Our model has a random-effect term to account for the fact that individuals were serially sampled. This person effect could account for systematic variation that was stable within an individual over the duration of the study (days 0–14), but would not account for changes in cell composition following vaccine administration. The corresponding paragraph in the manuscript has been corrected to reflect this.

*7) The authors use the delta r^2^ metric to identify eQTLs that differ between time points or in response to the vaccination. It seems that the authors could have performed the analysis using the delta of transcription(time point x)−transcription(baseline) as a trait. This normalizes to each individual’s starting point. Furthermore, a difference of r^2^ between time points can have multiple causes (as the authors point out). It seems to us that what they are most concerned with is cases where the effect size (beta) of a given eQTL differs between time points,* or *where the actual response (transcription time point x−transcription baseline) differs between individuals. The authors reasoning behind these choices should be made clearer*.

We attempted analyses with an approach similar to that proposed by the reviewers in the course of our work, but found that the approach that was ultimately chosen to explore the day differences was the most powerful. Specifically, utilizing a pairwise comparison (difference) between time points as the substrate for the eQTL analysis would lead to an increase in the technical variance of the phenotype, as the sum of two independent (technical) errors has twice the variance of an individual measurement. We agree with the reviewer that our interest is in investigating the source of changes in R^2^ explained on each day, and we analyzed this change in R^2^ to show that (a) in principle, R^2^ for the explanatory power of the genotype increases in the days after vaccination, and (b) this change is due to an increase in the magnitude of the additive effect, and not in a change in the dispersion of the errors across days.

*8) For the trans-eQTL analysis, the authors consider as genome-wide significant any SNP-transcript pair with p<5×10^-8^. Indeed for a GWAS of common variation, this may be appropriate for a single trait, but the authors should account for testing nearly 10K transcripts*.

In this study we performed only *cis*-eQTL analysis within an interval of 1 Mb centered on the reporter sequence. Positive associations in this analysis are most likely to result from direct allele-specific effects on transcription rate or transcript stability. A p-value threshold of 5 x 10^-8^ would, therefore, be conservative based on either number of SNPs or LD structure around the transcript. Future studies with much larger sample sizes might allow a full trans-eQTL analysis in which one would seek to identify genetic variants at other loci that indirectly affect transcript abundance measures. We have removed the term “genome-wide significance”, in reference to the p-value, from the relevant section of the Results.

*[After resubmission, the editors also asked for the following comments to be addressed prior to acceptance.*]

*A) The revision is not satisfactory for point 7, above, on the delta r^2^. The case is still not convincing because only 146 of the 541 cases were validated. It is not clear why the difference in r^2^ is necessarily interpreted as a significant change when only 30% of the cases are validated*.

We chose to highlight the observed phenomenon of a change in the magnitude of the genetic effect after the experimental stimulus (delta-Rg^2^) because it is evident from our data (as exemplified by the *NECAB2* locus in Figure 2) and it is a general property of the observed eQTLs (Figure 2). Importantly, this observation could not be made by cross-sectional study designs such as genome-wide association studies or single-time-point eQTL analyses.

The Reviewing editor has raised concern about the validation of the delta-Rg^2^ property in the two cohorts. We agree that the metric is not well defined in terms of the type I error, and that we should avoid the term ‘validation’ in this setting. The manuscript has been modified to reflect this change. However, our data shows that the delta-Rg^2^ is a stable property of SNP-transcript pairs in both cohorts. To more objectively illustrate this, we evaluated the Spearman's rank correlation of the delta-Rg^2^ between the two cohorts using all SNP-transcript pairs (Cor = 0.25, p < 2x10^-16^). This information has been added to the manuscript. To select a conservative set of candidate loci based on this phenomenon, we chose to impose a stringent intersection rule, requiring a threshold of the top 1% of the delta-Rg^2^ distribution in both cohorts, in addition to evidence of a strong *cis*- acting eQTL (genotype effect p < 5 × 10^-8^) in both cohorts. The manuscript was modified to clarify that the 146 SNP-transcript pairs that were selected for further analysis comprise the set that passed this strict intersection rule.

*B) The modelling as previously discussed in point 3 is somewhat weak and causality is not as well demonstrated, as the authors want to claim. The fact that the association with the residuals is lower than with the original values is not direct but indirect evidence that there are casual relationships. Also, how were the “random” SNPs chosen? Establishing a relevant null is important here. Also, direct causal models would help in establishing the relationships rather than the indirect approach the authors use*.

We agree with the Reviewing editor and, as stated above, spent considerable time reconsidering our approach to the evaluation of causal versus alternative association model comparisons.

Our work considered three distinct types of associations in the vaccine response data: genotype to expression (eQTL), expression to antibody response and genotype to antibody response. An important advantage of experimental designs such as ours is the opportunity to consider relationships between these distinct types of association. To respond to the reviewing editor's questions and to more formally evaluate alternative models, we have now adapted the approach recently published by Eric Schadt and colleagues (Millstein et al., 2009), which provides a statistical testing framework for so-called causal versus reactive model analyses. This work is related to ideas and methods developed by Judea Pearl and others, but more specifically applied to the context of eQTL data. Under this approach, a formal statistical hypothesis test is developed for evaluating three possible scenarios, referred to as independent, causal and reactive models.

This concept is now depicted in Figure 7 and described in the text of our paper. This formulation incorporates the three pairs of associations considered in our study: locus-to-expression, expression-to-trait, and locus-to-trait, together with a more explicit analysis of conditional independence between two of these variables given the third, such as the association (and change in association) between locus and trait conditional on gene expression, which is the focus of the causal model. Two separate p-values are generated, considering both the causal and reactive scenarios against the null model of independence. Details of our approach are described in the Methods section of the paper and the results of the analyses are presented in Figure 7 and described in the text. In keeping with the Reviewing editor's comments, and in an attempt to more globally evaluate the trend observed in our data, we performed a large-scale simulation under a null model that used permutation to sever the connection between eQTL and expression-to-trait associations. In each permutation, we selected a set of candidate SNP-transcript pairs that passed eQTL and expression-to-permuted-phenotype association. We examined the causal versus reactive p-value distributions for these simulation-based results. In no instance did we observe the trend towards the causal versus the reactive model that was observed in our data.

Finally, we performed a power analysis to examine the sample size that would be required to establish statistical significance under this methodology when effect sizes and errors are consistent with our data. These analyses suggest that a sample size of 200 – which corresponds to the combined sample size of our two cohorts – has 50–60% power to detect deviations from the independence model if they are present (Figure 7). This result suggests that our sample size, while giving us some opportunity to discriminate these alternatives, is still too small, and that a sample size at least twice as large would be needed for more clarity.